# Replacing meat with alternative plant-based products (RE-MAPs): protocol for a randomised controlled trial of a behavioural intervention to reduce meat consumption

Filippo Bianchi,[1] Paul Aveyard,[2] Nerys M Astbury,[1] Brian Cook,[2] Emma Cartwright,[3] Susan A Jebb[2]

¹Nuffield Department of Primary Care Health Science, University of Oxford, Oxford, UK
²Primary Care Health Sciences, University of Oxford, Oxford, UK
³Lee Kong Chain medical school, Nanyang Technological University, Singapore, Singapore

**Correspondence to**
Filippo Bianchi;
filippo.bianchi@phc.ox.ac.uk

## ABSTRACT

**Introduction** Reducing meat consumption could contribute towards preventing some chronic conditions and protecting the natural environment. This study will examine the effectiveness of a behavioural intervention to reduce meat consumption.

**Methods and analyses** Replacing meat with alternative plant-based product is a randomised controlled trial comparing a behavioural intervention to reduce meat consumption with a no intervention control condition. Eligible volunteers will be recruited from the general public through advertisement and randomised in a 1:1 ratio to receive no intervention or a 4-week intervention comprising the provision of free plant-based meat alternatives, written information on the health and environmental benefits of eating less meat, success stories of people who reduced their meat consumption and recipes. The primary outcome is the change in meat consumption at 4 weeks (T1) from baseline. Secondary and exploratory outcomes include changes in meat consumption at 8 weeks (T2) from baseline and changes from the baseline to both follow-up in other aspects of participants diet, putative psychosocial determinants of eating a low meat diet and of using meat substitutes and biomarkers of health risk, including blood lipid profiles, blood pressure, weight and body composition. Linear models will be employed to explore whether the changes in each of the aforementioned outcomes differ significantly between the control and intervention group. Qualitative interviews on a subsample of participants receiving the intervention will evaluate their experiences of the intervention and help to identify the mechanisms through which the intervention reduced meat consumption or the barriers preventing the intervention to aid this dietary transition.

**Ethics and dissemination** The trial has been granted ethical approval by the Medical Sciences Interdivisional Research Ethics Committee (IDREC) of the University of Oxford (Ref: R54329/RE001). All results originating from this study will be submitted for publication in scientific journals and presented at meetings and through the media.

**Trial registration number** ISRCTN13180635;Prerecruitment.

### Strengths and limitations of this study

► The first randomised controlled trial assessing the behavioural, nutritional, psychosocial and health impact of a 4-week intervention to reduce meat consumption through replacement with meat alternatives.

► Assessment of putative psychosocial determinants of meat and meat alternatives consumption will help to identify the active components of the intervention and will help inform future intervention development.

► Health risk outcomes will provide preliminary evidence on potential health implications of replacing meat with meat alternatives in the diet.

► Recruitment will occur among adult-only households within Oxford (UK), limiting the generalisability of the results.

► The study will only provide proof of principle for the short-term effectiveness of a behavioural intervention to reduce meat consumption and future work will be needed to translate these insights into longer term interventions in routine settings.

## INTRODUCTION

While meat is a source of important nutrients and can be part of a healthy diet,[1] red and processed meat consumption is also associated with an increased risk of developing some forms of cancer,[2–4] cardiovascular disease[5–8] and type 2 diabetes.[7 9–11] Furthermore, producing meat can negatively affect the natural environment and contribute to anthropogenic global warming,[12–14] which may also detrimentally affect human health.[15–18] Reducing meat consumption couldhelp to promote public health and protect the natural environment, but a recent report identified 'a remarkable lack of policies, initiatives or campaigns' designed to tackle the demand for meat.[19] This state of inaction is partly due to the scarcity of

evidence on the effectiveness of interventions to reduce meat consumption[19–23] warranting more experimental research to develop and evaluate such interventions. The rising availability of alternatives, such as textured vegetable proteins and mycoprotein-based alternatives,[24] could help to reduce meat consumption, as these products resemble meat in their gastronomic function, appearance and preparation. Nevertheless, uptake of meat alternatives remains low in many developed countries,[24–27] which might partly be due to a lack of familiarity with these foods.[25 28 29] Interventions increasing people's familiarity with meat alternatives could, therefore, help to overcome this barrier and, in turn, reduce meat consumption. A recent systematic review of experimental studies concluded that interventions that supplied plant-based alternatives were associated with reductions in meat consumption during, and several weeks after, the interventions.[23] Nevertheless, this evidence is based on small uncontrolled pre–post intervention studies[30 31] and more systematic evaluations of the behavioural impact of such interventions is warranted. Additionally, there is currently no evidence from randomised trials on the psychosocial and health consequences of interventions aiming at reducing meat consumption through the replacement with meat alternatives.

## Objectives

The primary aim of the replacing meat with alternative plant-based products (RE-MAP) trial is to examine the effectiveness of a behavioural intervention to reduce meat consumption compared with a no intervention control condition. Additionally, this study will evaluate the impact of the intervention on the consumption of other food groups, the nutritional composition of participants' diets, the putative psychosocial determinants of eating a low meat diet and of using plant-based meat alternatives and on biological markers of health risk, including blood lipid profiles, blood pressure, weight and body composition. This study also aims to qualitatively investigate participants' experiences of the intervention, the mechanisms through which the intervention reduced meat consumption, and/or the barriers preventing the intervention to aid this dietary transition.

## METHODS
### Study design and setting

The Re-MAP study will employ a two-arm parallel group individually randomised controlled trial (RCT) to evaluate a 4-week behavioural intervention to reduce meat consumption. The primary endpoint is defined as the change in average daily meat consumption at 4 weeks form baseline, assessed through self-reported 7 days food diaries. The study will be conducted in Oxford, UK.

### Recruitment

Participants will be recruited from the general population through advertisements in public buildings, newspapers,

mailing lists and social media. Individuals contacting the study team will receive a written information sheet summarising the study protocol. Individuals confirming their interest will be called by the recruiting member of the research team, who will summarise the study protocol and answer any outstanding question. The recruiting member of the research team will also screen individuals against the eligibility criteria and invite eligible individuals to attend an enrolment appointment.

### Eligibility criteria

Inclusion criteria:
A. Are ≥18 years old.
B. Self-report to eat meat regularly.
C. Belong to an adult-only household.
D. Are willing to try meat alternatives.
E. Own adequate food storing facilities.
F. Possess a device compatible with the requirements of the online food diary.
G. Provide informed consent.

Exclusion criteria:
A. Report they have relevant food allergies.
B. Report suffering from an eating disorder.
C. Report to be pregnant or plan pregnancy in the study period.
D. Belong to the same household as a previously enrolled participant.
E. Report consuming meat alternatives more than once a week on average.
F. Return baseline dietary records of insufficient quality for analysis.
G. The recruiting researcher deems the interested individual unable to adhere appropriately to the study protocol (eg, insufficient knowledge of the English language, planned absences from main residence during the course of the study, enrolled in other longitudinal dietary intervention study).

### Participant flow
#### Enrolment appointment

The enrolment appointment will take place on the university premises. During this appointment, an appropriately trained member of the research team will seek written informed consent (see online supplementary file 1), witnessing this by means of dated signature. After gaining informed consent, the enrolling member of the research team will set up participants' online food diaries to include six possible meal entries per day (breakfast, mid-morning, lunch, mid-afternoon, dinner and post-dinner) and to allow the research team to remotely access participants' food diaries by means of a password. The recruiting member of the research team will also train participants in how to appropriately use the online food diaries and estimate portion sizes.

#### Baseline

Following the enrolment appointment, participants will complete a 7-day food diary over the week leading up to

the following appointment, the baseline (T0). Participants not keeping sufficiently detailed diaries and those eating meat on less than five eating occasions over the week will be discontinued. At the baseline appointment, an appropriately trained member of the research team will collect participants' food diaries, ask participants to answer the baseline online questionnaire and measure participants blood lipids profile, blood pressure, weight and body composition. At the end of the baseline appointment, participants will be randomised to one of the two study conditions and will then follow the respective protocol for the next 4 weeks.

### Follow-up
Participants will be invited to attend a 4-week (T1) and an 8-week (T2) follow-up and to keep a 7-day food diary over the week leading up to each follow-up. During the follow-up appointments, a member of the research team will collect the respective food diary, ask participants to answer an online questionnaire and measure participants blood lipids profile, blood pressure, weight and body composition.

### Sample size
Due to lack of research studies directly comparable to ours, pragmatic considerations have guided the decision to terminate recruitment once a sample of at least 100 volunteers has completed the 4 weeks follow-up. A power analysis based on this pragmatically selected sample size suggests that 100 participants completing the primary outcome will allow detection of a medium effect size of d=0.6 with a power of 1-beta=0.84 and a two-tailed alpha criterion of 0.05.

### Randomisation and blinding
Participants' group allocation will be based on a computer-generated randomisation sequence, produced by an independent statistician. The randomisation sequence was designed to individually allocate participants to the intervention or control condition in a 1:1 ratio and to achieve a proportional gender balance in the two conditions through blocking and stratification by sex. The research team is blinded to the randomisation sequence and to its block sizes and sequence. Allocation will be revealed to the researcher performing the randomisation only after the first food diary has been returned. Due to the nature of the intervention, participants and some members of the research team cannot be blind to participants' group allocation. The members of the research team analysing the food diaries will be blind to the group allocation. Due to the nature of the outcomes, the risk of investigator bias will be low. To address the risk of social desirability bias in participants' reporting of foods intake and questionnaire responses, participants will be reminded during the enrolment visit and before each questionnaire that there are no right or wrong answers.

## Intervention and comparator
### Intervention
Re-MAP is a 4-week behavioural intervention, which aims to reduce meat consumption, defined as non-seafood meat products, among regular meat eaters. Following an analysis of the target behaviour, that is, a reduction in meat consumption, we included five psychosocial variables as the intervention's targets: attitudes, perceived behavioural control and subjective social norms of eating a low meat diet, as well as attachment to meat, and eating identities (eg, 'meat eater' or 'vegetarian'). We then selected four intervention functions from the Behaviour Change Wheel[32 33] with the aim of influencing these psychosocial variables[1]: environmental restructuring enacted through providing meat alternatives for 4 weeks,[2] training enacted through recipes,[3] education enacted through infographics on the health and environmental benefits of eating less meat, and[4] social modelling enacted through written vignettes outlining the story of people who reduced their meat consumption. These success stories were developed following an online patient and public involvement (PPI) activity. This PPI activity involved asking people who consciously reduced their consumption of meat to share their motives to do so, their strategies to enact this dietary transition, and the way they overcame the challenges associated with this transition. A logic model of the intervention is displayed in figure 1.

### Comparator
Participants in the control condition will receive no intervention. The template for intervention description and replication (TiDIER) checklist[34] for the Re-MAP intervention and the comparator is reported in table 1.

### Patients and public involvement
Following the development of the basic intervention structure, we held a discussion group with 10 members of the general public aiming to improve the acceptability and effectiveness of the RE-MAP intervention. We invited five meat eaters and five meat reducers to attend the discussion group, aiming to include people representing the target population of the intervention as well as people that successfully reduced their meat consumption. Public contributors were recruited using an established mailing list. The discussion group informed the development of each intervention component and of other aspects of the trial including:
► What type of meat alternatives to offer as part of the intervention.
► How to design the educational intervention components to be engaging and easily accessible to different publics.
► What language to use as part of the success stories vignettes and how to increase their relatability.
► What cookbooks and recipes to use as part of the intervention.

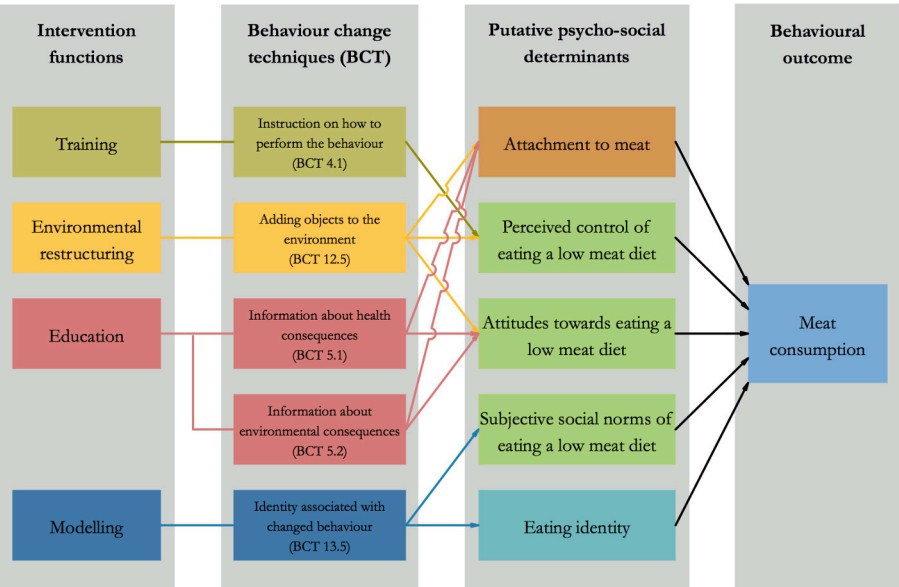

**Figure 1** Intervention logic model.

► The likely burden of trial participation and how to best compensate trial participants.

Contributors to the aforementioned public involvement activities will not be involved in other aspects of the trial implementation (such as recruitment) and will be asked not to enrol as trial participants, as they will have already reviewed much of the intervention material.

## Outcomes
### Primary outcome
► Change in mean daily grams of meat consumed between the baseline (T0) and the 4-week follow-up (T1).

### Secondary outcomes
► Change in mean daily grams of meat consumed between the baseline (T0) and the 8-week follow-up (T2).
► Change in the intention to eat a low meat diet between the baseline (T0) and both follow-up (T1, T2).
► Change in attachment to meat, eating identities and in attitudes, perceived behavioural control, and subjective social norm of eating a low meat diet between the baseline (T0) and both follow-up (T1, T2).

### Exploratory outcomes
► Change in participants' blood lipid profiles (total cholesterol, High-density lipoprotein (HDL) cholesterol, triglycerides, Low-density lipoprotein (LDL) cholesterol, non-HDL cholesterol, LDL:HDL cholesterol ratio) between the baseline (T0) and both follow-up (T1, T2).
► Change in systolic and diastolic blood pressure between the baseline (T0) and both follow-up (T1, T2).
► Change in participants' body mass index between the baseline (T0) and both follow-up (T1, T2).

► Change in participants' body fat percentage between the baseline (T0) and both follow-up (T1, T2).
► Change in the number of meals containing foods from other food groups between the baseline (T0) and both follow-up (T1, T2).
► Change in participants' mean daily energy, macronutrients and micronutrients intake between the baseline (T0) and both follow-up (T1, T2).
► Change in participants' intentions, attitudes, perceived behavioural control and subjective social norms of using meat alternatives between the baseline (T0) and both follow-up (T1, T2).
► Change in participants' desire for meat substitutes to be similar to meat between the baseline (T0) and both follow-up (T1, T2).

## Measurements
Table 2 provides a summary of the trial activities and of the measurement that will be collected at each stage of the trial.

### Sociodemographic characteristics
► At the baseline, participants will be asked to self-report on their age, sex, highest degree, household income, household composition, ethnicity and nationality.

### Psychological trait characteristics
► At the baseline, participants' trait food neophobia will be measured using a questionnaire scale adapted from Pliner and Hobden[35] including six items with a 7-point scale (disagree strongly—agree strongly).
► At the baseline, participants' self-control will also be assessed using a questionnaire scale adapted from Tangney[36] including eight items with a 7-point scale (disagree strongly—agree strongly).

**Table 1** TiDIER checklist describing the replacing meat with alternative plant-based product (Re-MAP) intervention and no-intervention comparator

| | Intervention | Comparator |
|---|---|---|
| Brief name | Re-MAP—a behavioural intervention to reduce meat consumption | No intervention |
| Why | Environmental restructuring: Meat alternatives will be provided for 1 month with the aim of enhancing attitudes towards and behavioural control of eating a low meat diet by making meat-free alternative easily available to participants. This intervention component also aims to reduce participants' attachment to meat. Participants will select from a range of commercially available meat alternatives including soy and other textured vegetable protein products (eg, soy sausages), plant-based and pulse-based products (eg, bean burgers), mycoprotein products (eg, mycoprotein steaks). Meat alternatives will be defined as meat-free products that fulfil the same gastronomic function as products that normally contain meat (eg, sausages, burgers, meatballs, steaks or mince).<br>Training: Recipes will be provided with the aim of enhancing participants' behavioural control of eating a low meat diet by enhancing their skills of preparing meat-free meals.<br>Education: Information leaflets about the health and environmental benefits of eating less meat will be provided to enhance participants' attitudes towards eating a low meat diet and to reduce participants' attachment to meat.<br>Social modelling: Written success stories of people who reduced their meat consumption will be provided to increase participants perceived social norm of eating a low meat diet and to promote the dietary identity of meat reducers, such as flexitarians. | N/A |
| What | Environmental restructuring: Participants will be provided with meat alternatives for 1 month, which they will be able to select from a printed catalogue of commercially available meat alternatives. Participants will be asked to select enough meat alternatives to have a meat-free product available on every occasion on which they would normally have meat for 2 weeks. Participants will be free to order enough foods to cater for themselves and other members of their household, if they wished to do so. The meat alternatives will be delivered to participants' homes by a food retailer on up to two occasions over the intervention month: the first delivery will be scheduled immediately after participants are allocated to the intervention condition. The second delivery will be scheduled 2 weeks after the randomisation for participants who wish to top up their stock of meat alternatives.<br>Training: A printed booklet containing 11 illustrated recipes of meat alternatives will be delivered immediately after participants are allocated to the intervention condition. These recipes will incorporate some of the meat alternatives used as part of this study. A second cookbook predominantly reporting on more general meat-free recipes (ie, not focussing on meat alternatives) will be provided during the fourth intervention week. All participants received the same recipes.<br>Education: Participants will receive eight printed pages of illustrated information on the health (four pages) and environmental implications (four pages) of eating less meat and two introduction pages and references delivered per post to their home over the course of the intervention month. The infographics were developed using publicly available information from peer-reviewed literature and relevant environmental or health organisations (eg, cancer research UK). Immediately after being allocated to the intervention condition, participants will receive an illustrated binder, which they will use to collect the information leaflets. The binder will include two pages of introductory information and the sources from which the information was drawn.<br>Success stories: Participants will receive three illustrated success stories vignettes delivered per post to their home during the last intervention week. The success stories will cover a range of different demographics (sex and age), different motives for eating less meat, and different strategies to transition to lower meat diets. The narratives will be about eating less meat rather than about ceasing to eat meat entirely. Participants will also receive a sheet on which they could report their own success story if they wish to do so. Participants will be asked to add this information to their illustrated binder. | N/A |
| Who | The lead researcher of this trial (FB) will deliver the intervention. An access database system will be used to schedule the deliveries of each intervention component ensuring that each intervention component will be delivered at the appropriate time for each participant. | N/A |
| How | The intervention consists in the delivery of the aforementioned materials. We will use the delivery services of one of UK's largest food retailers to purchase and deliver the meat alternatives to participants. We will use Royal Mail to send printed materials. The binder will be delivered to participants immediately after they are randomised to the intervention condition. | N/A |
| Where | N/A | N/A |
| Tailored | N/A | N/A |
| How well | We elected to use a single study account with the food retailer to schedule all the study deliveries, which will enable us to monitor the successful completion and receipt of each delivery. Due to the nature of the intervention, it will not be necessary to establish any other systems to monitor the fidelity of the intervention delivery. | N/A |

N/A, not available; TiDIER, template for intervention description and replication.

**Table 2** Schedule of measurements and trial activities

| | Visits | | | | |
| --- | --- | --- | --- | --- | --- |
| | Telephone screening | Enrolment visit | Baseline visit | 4-week follow-up | 8-week follow-up |
| **Enrolment** | | | | | |
| Eligibility screening | X | | | | |
| Informed consent | | X | | | |
| Randomisation | | | X | | |
| **Intervention** | | | | | |
| Replacing meat with alternative plant-based product | | | | | |
| **Control** | | | | | |
| Demographic and psychosocial traits | | | | | |
| Demographics | | | X | | |
| Food neophobia | | | X | | |
| Self-control scale | | | X | | |
| Dietary measurements | | | | | |
| Food diary | | | X | X | X |
| Retrospective eating questionnaire | | | X | X | X |
| Psychosocial variables | | | | | |
| Attitude towards eating a low meat diet and using meat alternatives | | | X | X | X |
| Perceived behavioural control of eating a low meat diet and using meat alternatives | | | X | X | X |
| Subjective social norm of eating a low meat diet and using meat alternatives | | | X | X | X |
| Intention to eat a low meat diet and to use meat alternatives | | | X | X | X |
| Attachment to meat | | | X | X | X |
| Eating identity | | | X | X | X |
| Desire for similarity between meat and meat alternatives | | | X | X | X |
| Biophysical outcomes | | | | | |
| Height | | | X | | |
| Weight | | | X | X | X |
| Body composition | | | X | X | X |
| Blood pressure | | | X | X | X |
| Blood lipids profile | | | X | X | X |
| Qualitative workstream | | | | | |
| Semistructured interviews | | | | | X |

### Dietary measurements

▶ Meat consumption will be measured in grams/day by disaggregating meat products recorded by participants on their 7 days food diaries. The daily average will exclude days in which energy intake was <1000 kcal, which are considered unlikely to represent habitual consumption.

▶ Average daily number of meals containing foods from other food groups will be measured counting the meals in participants' food diaries containing the food groups of interest, including:
 – Unprocessed pork meat.
 – Unprocessed red meat from ruminants.
 – Unprocessed poultry or game meat.
 – Processed meat.
 – Mycoprotein meat alternatives.
 – Soy-based meat alternatives or meat alternatives made of other textured vegetable protein.
 – Other meat alternatives (eg, bean burgers).
 – Milk and yoghurt.
 – Cheese.
 – Dairy-free milk and yoghurt alternatives.
 – Dairy-free cheese alternatives.
 – Fish and seafood.

- Eggs.
- Pulses other than those in meat alternatives.
- Vegetables other than those in meat alternatives.
- Starchy foods other than those in meat alternatives.
- Nuts and seeds other than those in meat alternatives.
- Fruit.
- Savoury and sweet snacks.
- Soft drinks.
- Alcoholic drinks.

A retrospective eating questionnaire will also ask participants to recall the number of eating occasions on which they had the foods listed above over the week of their food diary. This questionnaire will only be used in sensitivity analyses.

► The daily average energy intake and nutritional composition of participants' diets will be measured using data from the online food diary.

### Psychosocial variables

► Attachment to meat will be measured using the meat attachment questionnaire.[37]

► Eating identities will be self-reported by participants among meat eater, omnivore, flexitarian, pescatarian, vegetarian, vegan or 'other'.

► Attitudes, subjective social norms and perceived behavioural control to eat a low meat diet and to use meat alternatives will be, respectively, assessed with three questionnaire items constructed following Francis et al[38] on a 7-point scale (disagree strongly—agree strongly).

► Intentions to eat a low meat diet and to use meat alternatives will be assessed using a single questionnaire item on a 7-point scale (disagree strongly—agree strongly).

► Desire for similarity between meat and meat alternatives will be assessed using 11 questionnaire items with a 7-point scale (disagree strongly—agree strongly) adapted from Hoek et al.[25]

### Physical measures

► Blood lipids profiles (total cholesterol, HDL cholesterol, triglycerides, LDL cholesterol, non-HDL cholesterol, LDL:HDL cholesterol ratio) will be measured using Alere Cholestech LDX.

► Height will be measured to the nearest 0.1 cm using a stadiometer.

► Weight and body composition will be measured using an electronic scale (SC-240 MA, Tanita, Japan), which records the proportion of body fat using bioelectrical impedance. Weight was recorded to the nearest 0.1 kg.

► Seated blood pressure will be measured as the average of the second and third reading of three seated readings

### Retention

We will use reminder text messages to increase attendance to each of the four study appointments. Additionally, participants will receive financial compensation for partaking in each of the three assessment visits. Participants will have the right to withdraw from the study at any time. The principal investigator will have the right to discontinue participants' involvement in the study when they become ineligible and/or when significant protocol deviations occur. The data of participants who withdraw will be kept and might be used in exploratory and sensitivity analyses, unless the participant requests for the data to be deleted.

### Adverse events

Any study-related adverse event will be reported to the research ethics committee in accordance to Good Clinical Practice (GCP). All study-related adverse events will be included in the final trial report.

### Data management

Data will be entered by a trained member of the research team and stored in an OpenClinica database that was specifically developed for this trial. The database will feature ranges and validation checks to promote reliability in the data entry process. Data recording and storage will run in accordance with GCP.

### Statistical analyses

We will employ linear models to investigate whether changes in meat consumption between the baseline and both follow-up differ significantly between the intervention and the control group. Our main analysis will employ unadjusted models and only include data from participants completing the relevant follow-up. Sensitivity analysis will be performed with a baseline observation carried forward assumption for missing data and adjusting for baseline variables. The intervention effect will be reported with 95% CI and p values. A two-tailed criterion p value of alpha=0.05 will be used to assess the statistical significance of the results. The same procedure will be employed to assess whether changes in the other prespecified dietary, nutritional, psychosocial and biophysical outcomes between the baseline and both follow-up differ significantly between the control and the intervention group. Detailed main, subgroup and sensitivity analyses plans will be finalised before conducting any specific outcome analysis. No interim analysis is planned.

### Qualitative study

After the 8-week follow-up, a subsample of participants receiving the intervention will be invited to take part in a semistructured interview. This qualitative study aims to understand participants' experiences of the intervention, the mechanisms through which the intervention helped reducing meat consumption or the barriers preventing the intervention to aid this transition. The semistructured interviews will follow a discussion guide while also remaining sensitive to unsolicited themes. The interview will set the context by asking participants to elaborate on their motivation to volunteer for the trial and on their thoughts and feelings towards reducing meat consumption prior to enrolling into the study. Participants will then

be encouraged to elaborate on the mechanisms through which they felt that the intervention helped them eat less meat or the barriers preventing the intervention to do so. In doing so, participants will be prompted to think about the intervention in its entirety as well as about each individual intervention component. Participants will be encouraged to elaborate on their perceived ability and motivation to maintain a lower consumption of meat after the intervention period and beyond the context of the study. Whenever possible we will use open questions to encourage participants to elaborate on their thoughts and feelings freely and in depth. We aim to avoid questions of evaluative nature to minimise the risk of social desirability bias. We anticipate interviewing 20 participants, however, sampling will be extended should new themes emerge during the interviewing process. We will employ a purposeful sampling technique aiming to achieve a sex balance. Participants will be free to decide whether or not to be interviewed. No additional compensation will be offered to participants agreeing to be interviewed. Qualitative interviews will be conducted in person and transcribed verbatim. Transcriptions will be analysed using NVivo and employing a data-driven thematic analysis to identify codes and to group these codes into broader themes.

## Trial steering committee

The principal investigator will be responsible for the project coordination and the senior investigators will oversee the operational aspects of the trial. The authors of this protocol will form the trial management group (TMG), which will regularly monitor the study implementation, as well as the data generation, documentation and reporting. All members of the TMG are trained in GCP and will take appropriate actions to safeguard participants and the quality of the trial. Access to data will be granted to appropriate members of the research team and to authorised representatives from the host institution to monitor and/or audit the study and ensure compliance with regulations.

## Ethics and dissemination

The investigators will ensure that this study is conducted in accordance with the principles of the Declaration of Helsinki, with relevant institutional regulations, with GCP and General Data Protection Regulations (GDPR). This study was reviewed and received ethical approval by the Medical Sciences Interdivisional Research Ethics Committee of the University of Oxford (R54329/RE001). Substantial planned changes to the protocol, an end of study notification, and a final report will be submitted to the aforementioned research ethics committee. The results of this RCT will be reported following the Consolidated Standards of Reporting Trials guidelines[39] and submitted for publication to scientific journals, regardless of the outcome. Authorship will be determined in accordance with the International Committee of Medical Journal Editors (ICMJE) guidelines. Contributors of other

parties and funding will be acknowledged. Results will also be presented at national and international conferences and disseminated through established networks. A lay summary will be distributed through an established newsletter to which participants can subscribe on their last study appointment.

**Acknowledgements**  We thank all the PPI contributors for having helped us develop the RE-MAP intervention. We thank Lynne Maddocks for her assistance in forming the PPI panel for this study. We thank Lucy Eldridge for her support in developing the study database. We thank Jason Oke for his assistance in developing the randomisation sequence. We thank Alexa Hayley and Bernhard Haring for their comments on previous versions of this manuscript.

**Contributors**  All authors have been involved in shaping each stage of this research protocol. FB has written this protocol and developed the intervention and led on the study design. FB, SAJ and PA have designed the study. FB and NMA have developed the trial management system. NMA, BC and EC have contributed in designing this research and the intervention.

**Funding**  This research is funded by the Wellcome Trust, Our Planet Our Health programme (Livestock, Environment and People—LEAP), award number 205212/Z/16/Z. FB's time on this project is funded by the Medical Research Council (MRC), Green Templeton College Oxford and the National Institute for Health Research (NIHR) School for Primary Care Research (SPCR). EC's and BC's time on this project is funded by the Wellcome Trust, Our Planet Our Health programme (Livestock, Environment and People - LEAP), award number 205212/Z/16/Z. NMA, PA and SAJ are supported by the NIHR Oxford Biomedical Research Centre and Collaboration for Leadership in Applied Health Research and Care Oxford at Oxford Health NHS Foundation Trust. PA and SAJ are NIHR Senior Investigators.

**Competing interests**  None declared.

**Patient consent for publication**  Not required.

**Ethics approval**  This study was reviewed and received ethical approval by the Medical Sciences Interdivisional Research Ethics Committee of the University of Oxford (R54329/RE001).

**Provenance and peer review**  Not commissioned; externally peer reviewed.

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
