## [Reviewer comments · BMJ Open]

ARTICLE DETAILS

TITLE (PROVISIONAL)	Replacing meat with alternative plant-based products (RE-MAPs): Protocol for a randomized controlled trial of a behavioural intervention to reduce meat consumption
AUTHORS	Bianchi, Filippo; Aveyard, Paul; Astbury, Nerys; Cook, Brian; Cartwright, Emma; Jebb, Susan

VERSION 1 - REVIEW

REVIEWER	Dr Alexa Hayley, Lecturer Deakin University, Australia
REVIEW RETURNED	30-Oct-2018

GENERAL COMMENTS	Dear Editor and Authors, Thank you for the opportunity to review this manuscript (Manuscript ID bmjopen-2018-027016) describing a protocol for an RCT intervention on reducing meat consumption. The authors present a well-written and interesting protocol, with an admirable intervention objective. Overall the protocol itself is sound and comprehensive, but there are some areas requiring clarity. Please see my comments/suggestions below: P3 line 19 – A clear definition and examples of meat-alternatives should be provided. This is explained in a broad way in lines 16-19, but should be followed with specific product examples by type, if not brand, e.g., are you referring just to ‘faux meats’ such as textured vegetable protein minces, gluten steaks/sausages, and mycoprotein? Or to plant-based products that are not necessarily protein-heavy but resemble burger patties, sausages, etc? Or to other protein-heavy plant-based products substituted in cuisines such as tofu and tempeh? To whole lentils, pulses? There are a variety of ‘meat alternatives’, all with different pros/cons to taste, texture, appearance, nutritional value, cost, and suitability to cuisine type. On P6 the authors propose to reduce ‘non seafood’ meat consumption – why isn’t seafood defined as a meat in the protocol? Does this mean that seafood can be treated as a ‘meat alternative’ by participants in the intervention condition? A clearer definition for terms and justification for what is/isn’t relevant is needed. P3 line 28 – include a comma after ‘weeks after’: “A recent systematic review of experimental studies concluded that
--

interventions that supplied plant-based alternatives were associated with reductions in meat consumption during, and several weeks after, the interventions (25).”

P3 line 46-50 – improve flow of sentence, for example: “This study also aims to qualitatively investigate participants’ experiences of the intervention, the mechanisms through which the intervention reduced meat consumption, and/or the barriers preventing the intervention to aid this dietary transition.”

P4 – For the inclusion/exclusion criteria, consider including additional screening items for whether others in the household do/don't regularly eat meat, and regarding the current economic status/circumstances of the applicant/participant (e.g., during the past month, have they experienced significant financial hardship impacting their ability to purchase/access sufficient groceries or other meals?). Including these items may allow you screen out those who are seeking access to meat-alternative products for others in their household, or access to free meals more generally (not genuine participants in both instances).

P4 – also problematic for inclusion/exclusion criteria, and the intervention more broadly, is how ‘meat-alternatives’ will be defined to potential participants during recruitment and the types of products that will be provided.

P4 – Within 'recruitment', it sounds as though quota sampling should be specified, since a 1:1 ratio is sought for control vs intervention regarding gender.

P6 – no need to repeat full description of Re-MAP acronym here; this was previously defined on P3 line 38.

P6 line 17 –Provide a definition/explanation for ‘eating identities’.

Page 7 line 6 – I agree with using a focus group of meat eaters/non meat eaters to assist with intervention materials, however I'd like to know more about how these materials were designed. Perhaps supplementary materials describing the intervention materials development could be provided:

- Were specific message-framing techniques employed based on past research/theory?
- Were recipes/cookbooks relevant to the types of meat alternatives offered?
- Was the likely varied cooking skill and cuisine preference of potential participants considered when choosing recipes/cookbooks?
- Were the awareness/education materials developed as independent assessments using peer reviewed literature, or drawn from social organisations likely familiar to participants or bearing some degree of authority?
- Regarding the ‘social modelling’ narratives, were a variety of diet/eating identities represented? Was consideration given to the stigmatising potential of including a vegan narrative?

Page 8 line 25 - typo: immediately ‘after’ they were randomised.

Page 8 line 37 – as the primary outcome being assessed in meat in grams consumed, and the participant must keep a food diary, consider providing or encouraging the purchase of a small electronic kitchen scale as part of the intervention, otherwise ‘grams’ may be guesstimates.

	Page 9 line 12-15: 'Change in the number of meals containing foods from other food groups between the baseline (T0) and both follow up (T1, T2)' sounds a bit vague- clarify briefly what is meant by 'other food groups' here (although I see this is described in detail on page 10). General comments: The protocol leaves ambiguous the types of meat alternatives participants can choose from, and the impact this likely variety will have on consequent/continued use. Participants' prior knowledge of meat alternatives, and the types of recipes/cookbooks provided, may influence the types of meat alternative products participants choose to order and also impact their expectations about their taste/texture (either positively or negatively). The intervention includes a number of confounds specific to individuals and to the breadth of meat-alternative products available; these confounds and limitations should be clearly acknowledged in the protocol, and some explanation given for how they will not impact the overall objective of the study outcomes and meaningfulness. The tense in the protocol changes at times; e.g., P5, both future and perfect present tense are used in the section 'Randomisation and blinding'. On P6 line 13 and line 54, past tense is used as though the intervention has already been conducted. Be consistent in how the protocol is described – as either something that will be enacted in the future, something that is currently being enacted, OR something that has already been enacted. I look forward to seeing this and associated results in print. Kind regards.
--	---

REVIEWER	Bernhard Haring University of Würzburg Germany
REVIEW RETURNED	03-Dec-2018

GENERAL COMMENTS	The manuscript 'Replacing meat with alternative plant-based products' provides a protocol for a randomized controlled trial of a behavioural intervention to reduce meat consumption. It is applaudable that the authors try to undertake such an endeavour since it is an important topic with great public health impact. I do have several suggestions: 1) Please be more specific with the wording (e.g. title): I assume, the authors focus on 'red' meat consumption only ? 2) Introduction: Please be more balanced, mention the issue that red meat consumption has not been observed to be entirely negativ, e.g. Am J Clin Nutr. 2012 Jan;95(1):9-16 or J Hum Hypertens. 2014 Oct;28(10):600-5. On the other hand, please provide further evidence on meat consumptions and its associations: The reviews by Micha et al. are from 2012 - quite a lot has been published since then, e.g. JAMA Intern Med. 2016 Oct 1;176(10):1453-1463, JAMA Intern Med. 2013 Jul 22;173(14):1328-35, etc. Please be more specific what type of meat you want to replace. It may be more useful and easier to identify food items such as burgers, ham consumption, etc. that you want to eliminate.
---

3) Intervention: It is not entirely clear to this reviewer how the authors want to replace (red) meat (certain food groups?). This is of great importance since replacing red meat products with another unhealthy food item may not be the best solution. Do the authors want to promote a specific dietary pattern (DASH, Mediterranean Diet, etc.) ? It seems that the whole study may ultimately end up in changing the dietary behavior (from e.g. a typical western diet to a e.g. DASH diet or any other healthy dietary pattern)? It has to be clear what type of products should be replaced and what food items should be promoted. Specifically, you may end up analyzing the diets of your study population and identify the specific food items that you want to eliminate and specify the food item that you want to promote (e.g. breakfast ham to be replaced by etc.)

4) Methods: Will the RCT be isocaloric ? The authors need to address this issue - it assume, it is not supposed to be a weight loss trial. Furthermore, the authors have to control for physical activity throughout the study. Yes, it is great to promote a healthy behaviour overall, however, the authors just want to analyze the effects of meat reduction in this trial.

5) Intervention: Will participants receive financial compensation - replacing meat may be costly. Household income and education of study participants may play a significant role as I assume this is not a feeding trial.

6) Intervention: The authors want to analyze explanatory variables such as BP, lipids, etc. Would it make sense to conduct a randomized, two-period, crossover study?

7) Intervention: Are there any safety concerns - will a dietician follow the participants? Protein consumption is always debated as a measure to prevent frailty in elderly individuals. Do the authors have any thoughts about this issue.

8) Intervention: How will the authors assess adherence to the dietary recommendation during the trial? Do you plan to study long-term adherence after the trial as well, as most dietary interventions show that after 6 months people tend to bounce back to old behavior.

9) Exploratory outcomes: potentially consider urinary samples (sodium, etc.) and stool analyses (e.g. change in microbioma).

10) Please provide a diagram/flow chart on enrolment and participant randomization and on the intervention.

11) Please reconsider the statistical methods and potential power analyses (e.g. for the secondary outcomes) as there is some literature on the association of reduction of red meat intake with various outcomes available. This is because several trials on promoting a healthy dietary pattern reduces meat intake as part of their Intervention.

12) Minor: The authors state: 'A recent systematic review of experimental studies concluded that interventions that supplied plant-based alternatives were associated with reductions in meat consumption during, and several...'. As a reference (25), they use a manuscript that has only be submitted - please provide a citation that has been already published.

13) Can you discuss potential limitations of this trial: Selection bias (recruitment), generalizability, blinding, etc. in more detail (own paragraph/subheading).

VERSION 1 – AUTHOR RESPONSE

Reviewer: 1

Reviewer Name: Dr Alexa Hayley, Lecturer, Institution and Country: Deakin University, Australia,
Please state any competing interests or state 'None declared': None declared

Dear Editor and Authors,

Thank you for the opportunity to review this manuscript (Manuscript ID bmjopen-2018-027016) describing a protocol for an RCT intervention on reducing meat consumption. The authors present a well-written and interesting protocol, with an admirable intervention objective. Overall the protocol itself is sound and comprehensive, but there are some areas requiring clarity. Please see my comments/suggestions below:

Thank you for appreciating the importance of this protocol and for your constructive comments. Below we explain how we changed our manuscript following your suggestions or why we think it is might not be appropriate or feasible to do so.

P3 line 19 – A clear definition and examples of meat-alternatives should be provided. This is explained in a broad way in lines 16-19, but should be followed with specific product examples by type, if not brand, e.g., are you referring just to 'faux meats' such as textured vegetable protein minces, gluten steaks/sausages, and mycoprotein? Or to plant-based products that are not necessarily protein-heavy but resemble burger patties, sausages, etc? Or to other protein-heavy plant-based products substituted in cuisines such as tofu and tempeh? To whole lentils, pulses? There are a variety of 'meat alternatives', all with different pros/cons to taste, texture, appearance, nutritional value, cost, and suitability to cuisine type. On P6 the authors propose to reduce 'non seafood' meat consumption – why isn't seafood defined as a meat in the protocol? Does this mean that seafood can be treated as a 'meat alternative' by participants in the intervention condition? A clearer definition for terms and justification for what is/isn't relevant is needed.

As suggested, we added specific examples of meat-alternatives to the introduction of our protocol (P.3, L.73-74) and more details about our definition of meat-alternatives in the section pertaining to the intervention (P.07, Table 01).

As fish consumption has different implications both for health and the natural environment, in this study we elected to only focus on reducing non-seafood meat. Thus, we neither discouraged fish consumption nor suggested it as an alternative to meat.

P3 line 28 – include a comma after 'weeks after': "A recent systematic review of experimental studies concluded that interventions that supplied plant-based alternatives were associated with reductions in meat consumption during, and several weeks after, the interventions (25)."

Thank you. We made this amendment. (P.3, L.82)

P3 line 46-50 – improve flow of sentence, for example: "This study also aims to qualitatively investigate participants' experiences of the intervention, the mechanisms through which the intervention reduced meat consumption, and/or the barriers preventing the intervention to aid this dietary transition."

Thank you. We made this amendment. (P.3, L.98-101)

P4 – For the inclusion/exclusion criteria, consider including additional screening items for whether others in the household do/don't regularly eat meat, and regarding the current economic status/circumstances of the applicant/participant (e.g., during the past month, have they experienced significant financial hardship impacting their ability to purchase/access sufficient groceries or other

meals?). Including these items may allow you screen out those who are seeking access to meat-alternative products for others in their household, or access to free meals more generally (not genuine participants in both instances).

As this is the first trial of this type - we elected not to put too many limits on recruitment. Should this intervention show some promise, future research might explore more specifically how to intervene among people from different socioeconomic background or among people from different households. The qualitative interviews of our study might provide important insights to inform how the aforementioned tailoring can be optimized.

P4 – also problematic for inclusion/exclusion criteria, and the intervention more broadly, is how ‘meat-alternatives’ will be defined to potential participants during recruitment and the types of products that will be provided.

The advertisement flyer is minimalistic and only states ‘meat-alternatives’. We explain the details of the study procedure at the phone to all individuals who express an interest in the trial. Interested individuals are told that they are free to select from a catalogue of commercially available meat substitutes, which we describe as ‘meat-free foods that have the same function of products that normally contain meat’. At the phone we also provide examples (e.g. meat-free sausages, burgers, meatballs, steaks, or mince) and we mention some well-known brands of meat-alternatives sold in the UK. We added more details pertaining to the definition of meat-alternatives in our revision (P.07, Table 01).

P4 – Within 'recruitment', it sounds as though quota sampling should be specified, since a 1:1 ratio is sought for control vs intervention regarding gender.

We included a stratification mechanism within the randomisation system, so that quotas will not be required to achieve a proportional sex balance between the two groups.

P6 – no need to repeat full description of Re-MAP acronym here; this was previously defined on P3 line 38.

Thank you. We made this amendment.

P6 line 17 –Provide a definition/explanation for ‘eating identities’.

We added some examples to clarify this in the protocol. (P.6, L. 21)

Page 7 line 6 – I agree with using a focus group of meat eaters/non meat eaters to assist with intervention materials, however I'd like to know more about how these materials were designed. Perhaps supplementary materials describing the intervention materials development could be provided:

- Were specific message-framing techniques employed based on past research/theory?
- Were recipes/cookbooks relevant to the types of meat alternatives offered?
- Was the likely varied cooking skill and cuisine preference of potential participants considered when choosing recipes/cookbooks?
- Were the awareness/education materials developed as independent assessments using peer reviewed literature, or drawn from social organisations likely familiar to participants or bearing some degree of authority?
- Regarding the ‘social modelling’ narratives, were a variety of diet/eating identities represented? Was consideration given to the stigmatising potential of including a vegan narrative?

Thank you for this comment. We have augmented our intervention description to provide more information about the questions you raise (P.7-8, Table 1).

Page 8 line 25 - typo: immediately 'after' they were randomised.

Thank you. We have amended this (P.8, Table 1).

Page 8 line 37 – as the primary outcome being assessed in meat in grams consumed, and the participant must keep a food diary, consider providing or encouraging the purchase of a small electronic kitchen scale as part of the intervention, otherwise 'grams' may be guestimates.

We decided against using scales because of feasibility issues and because we thought that asking participants to weigh their foods might detrimentally influence retention. Instead we decided to train participants in how to appropriately estimate and record food quantities during the enrolment visit. We have added a note mentioning this training session on P.5, L 10-12.

Page 9 line 12-15: 'Change in the number of meals containing foods from other food groups between the baseline (T0) and both follow up (T1, T2)' sounds a bit vague- clarify briefly what is meant by 'other food groups' here (although I see this is described in detail on page 10).

Thank you. To avoid repetition we have listed these just once, however we are happy to repeat these at the editors discretion.

General comments:

The protocol leaves ambiguous the types of meat alternatives participants can choose from, and the impact this likely variety will have on consequent/continued use. Participants' prior knowledge of meat alternatives, and the types of recipes/cookbooks provided, may influence the types of meat alternative products participants choose to order and also impact their expectations about their taste/texture (either positively or negatively). The intervention includes a number of confounds specific to individuals and to the breadth of meat-alternative products available; these confounds and limitations should be clearly acknowledged in the protocol, and some explanation given for how they will not impact the overall objective of the study outcomes and meaningfulness.

The tense in the protocol changes at times; e.g., P5, both future and perfect present tense are used in the section 'Randomisation and blinding'. On P6 line 13 and line 54, past tense is used as though the intervention has already been conducted. Be consistent in how the protocol is described – as either something that will be enacted in the future, something that is currently being enacted, OR something that has already been enacted.

I look forward to seeing this and associated results in print.

Kind regards.

This is the first trial evaluating the effectiveness of a complex behavioural intervention featuring meat-alternatives, infographics about the health and environmental implications of meat consumption, success stories of meat-reducers, and recipes. The primary outcome will explore whether and how this intervention – as a whole – changes participants' meat consumption. In our revision we more clearly describe each component of the intervention and particularly the type of products that we defined as meat-alternatives. We have amended our manuscript to consistently use the future tense. We thank you for your helpful comments and acknowledged your contribution in our paper.

Reviewer: 2

Reviewer Name: Bernhard Haring, Institution and Country: University of Würzburg, Germany, Please state any competing interests or state 'None declared': None declared

The manuscript 'Replacing meat with alternative plant-based products' provides a protocol for a randomized controlled trial of a behavioural intervention to reduce meat consumption. It is applaudable that the authors try to undertake such an endeavour since it is an important topic with great public health impact.

Thank you for your time to provide valuable feedback and for recognizing the public health importance of this research. We have acknowledged your contribution in our paper.

I do have several suggestions:

1) Please be more specific with the wording (e.g. title): I assume, the authors focus on 'red' meat consumption only?

This intervention was developed to address not only the public health impact of eating meat but also the environmental impact of producing meat. As such, the primary objective is to investigate whether the intervention reduces meat consumption in general. However, our outcome will allow for subgroup analysis investigating which types of meat changed the most as a result of the intervention.

2) Introduction: Please be more balanced, mention the issue that red meat consumption has not been observed to be entirely negative, e.g. *Am J Clin Nutr.* 2012 Jan;95(1):9-16 or *J Hum Hypertens.* 2014 Oct;28(10):600-5. On the other hand, please provide further evidence on meat consumption and its associations: The reviews by Micha et al. are from 2012 - quite a lot has been published since then, e.g. *JAMA Intern Med.* 2016 Oct 1;176(10):1453-1463, *JAMA Intern Med.* 2013 Jul 22;173(14):1328-35, etc. Please be more specific what type of meat you want to replace. It may be more useful and easier to identify food items such as burgers, ham consumption, etc. that you want to eliminate.

Thank you for this suggestion. We mention that meat is a source of important nutrients and, following your suggestion, we augmented the introduction to more explicitly state that meat can be integrated as part of a healthy diet (P.3, L.62). We also added some of the references you suggested to strengthen the evidence base underlying the introduction (P.3, L.64-65).

3) Intervention: It is not entirely clear to this reviewer how the authors want to replace (red) meat (certain food groups?). This is of great importance since replacing red meat products with another unhealthy food item may not be the best solution. Do the authors want to promote a specific dietary pattern (DASH, Mediterranean Diet, etc.)? It seems that the whole study may ultimately end up in changing the dietary behavior (from e.g. a typical western diet to a e.g. DASH diet or any other healthy dietary pattern)? It has to be clear what type of products should be replaced and what food items should be promoted. Specifically, you may end up analyzing the diets of your study population and identify the specific food items that you want to eliminate and specify the food item that you want to promote (e.g. breakfast ham to be replaced by etc.)

This study does not primarily aim to understand the impact of a specific diet on health. Instead we aim to investigate whether and how a multicomponent behavioural intervention changes participants' meat consumption. As such, we will not try to prescribe a shift towards a specific dietary pattern but instead capture whether and how the exposure to the intervention changes participants' meat consumption.

4) Methods: Will the RCT be isocaloric? The authors need to address this issue - it is assumed, it is not supposed to be a weight loss trial. Furthermore, the authors have to control for physical activity throughout the study. Yes, it is great to promote a healthy behaviour overall, however, the authors just want to analyze the effects of meat reduction in this trial.

As this is a behavioural intervention, we will not prescribe participants to follow a specific diet (e.g. a diet that is low in meat but isocaloric to their current dietary regimen). Instead we will investigate whether and how the intervention changes participants' eating behaviours.

In terms of calories, we will conduct exploratory analysis examining whether the intervention changed participants' energy intakes (See list of 'exploratory outcomes' on P.9).

5) Intervention: Will participants receive financial compensation - replacing meat may be costly. Household income and education of study participants may play a significant role as I assume this is not a feeding trial.

On P.12, L.21 we outline that participants in both groups will receive a financial compensation for their participation. Participants in the intervention condition will receive free meat-alternatives for one month, as outlined in the TIDIER intervention description on P.7, Table 1. We will not prescribe to participants to follow any specific diet. As such, if costs prevent them to shift towards a low meat diet (during or after the intervention) this will be captured in the trial's outcome.

6) Intervention: The authors want to analyze explanatory variables such as BP, lipids, etc. Would it make sense to conduct a randomized, two-period, crossover study?

The behavioural intervention was designed to promote long lasting self-motivated changes in participants' meat consumption that might carry on after the intervention completion - hence the follow up one month after the intervention completion. As such, we elected to conduct a parallel arm RCT rather than a cross-over study.

7) Intervention: Are there any safety concerns - will a dietician follow the participants? Protein consumption is always debated as a measure to prevent frailty in elderly individuals. Do the authors have any thoughts about this issue.

Thank you for your comment. The study team includes a general practitioner to whom participants can express any concerns they might have in relation to their diet and health at any stage of the trial. Additionally, participants were informed that they were enrolling in a behavioural study and that they would not be required to adhere to any specific dietary regimen.

8) Intervention: How will the authors assess adherence to the dietary recommendation during the trial? Do you plan to study long-term adherence after the trial as well, as most dietary interventions show that after 6 months people tend to bounce back to old behavior.

This is an experimental behavioural intervention study in which we investigate whether and how providing meat-alternatives, infographics on the health and environmental implications of eating meat, success stories of meat reducers, and recipes changes participants' consumption of meat during the intervention and four weeks after the intervention completion. As such, we do not ask participants to follow a specific diet and so there are no recommendations to which participants are asked to adhere. Whether participants' diet changes as a result of the intervention is the outcome – rather than a prerequisite - of the study. If the results of this first trial show promise, future research might explore the longer terms effects at 6 months or longer.

9) Exploratory outcomes: potentially consider urinary samples (sodium, etc.) and stool analyses (e.g. change in microbioma).

Thank you. We did consider stools and urinary samples. However after having piloted the procedures required for collecting, storing, transporting, and analyzing these samples we concluded that this would pose an excessive burden on participants and would not be feasible for our study.

10) Please provide a diagram/flow chart on enrollement and participant randomization and on the intervention.

Table 2 provides the flow chart of participants and the measures to be taken at each stage of the protocol. We have added an additional consort diagram to the submission.

11) Please reconsider the statistical methods and potential power analyses (e.g. for the secondary outcomes) as there is some literature on the association of reduction of red meat intake with various outcomes available. This is because several trials on promoting a healthy dietary pattern reduces meat intake as part of their Intervention.

Our sample size is largely based on practical considerations and on our primary outcome, as we outline on P.5, L.35-40. We are not aware of a behavioural intervention comparable to ours, which we could use to compute an informed power calculations.

12) Minor: The authors state: 'A recent systematic review of experimental studies concluded that interventions that supplied plant-based alternatives were associated with reductions in meat consumption during, and several...'. As a reference (25), they use a manuscript that has only be submitted - please provide a citation that has been already published.

Thank you. We have changed this to a published citation in our revision (P.3, L.82).

13) Can you discuss potential limitations of this trial: Selection bias (recruitment), generalizability, blinding, etc. in more detail (own paragraph/subheading).

We are keen to retain the headings from the SPIRIT guidelines. However, following your suggestions, we have added some considerations pertaining to blinding in the methods section (P.6, L. 6-12). We outline issues pertaining to the generalizability of the results on P.2, L. 54-55.

VERSION 2 – REVIEW

REVIEWER	Alexa Hayley Deakin University
REVIEW RETURNED	25-Feb-2019

GENERAL COMMENTS	Dear Authors, I am very pleased with the manuscript revisions and look forward to seeing the study's results. My concern remains regarding categorising 'pescatarian' as a non-meat eating category, however I do appreciate your explanation regarding the neutral approach to fish/seafood consumption in the intervention. If the decision to combine these diet identities is due to pragmatic reasons (i.e., low cell count for planned analyses), then I understand this decision however it should be noted as a limitation to the study. I advise that any reports describing intervention results provide a strong health-focused and design-focused justification for the decision to group pescatarians with vegetarians and vegans. This decision may impair the value of the study for some researchers investigating interventions for meat-
---

	reduced and meat-abstaining diets due to the conflation of diet identities/behaviours. All the best with your research. Kind regards, Alexa
--	--

REVIEWER	Bernhard Haring University of Wuerzburg, Germany
REVIEW RETURNED	15-Jan-2019

GENERAL COMMENTS	The reviewer completed the checklist but made no further comments.
--

VERSION 2 – AUTHOR RESPONSE

Review 2: N/A

Reviewer 1: I am very pleased with the manuscript revisions and look forward to seeing the study's results. My concern remains regarding categorising 'pescatarian' as a non-meat eating category, however I do appreciate your explanation regarding the neutral approach to fish/seafood consumption in the intervention. If the decision to combine these diet identities is due to pragmatic reasons (i.e., low cell count for planned analyses), then I understand this decision however it should be noted as a limitation to the study. I advise that any reports describing intervention results provide a strong health-focused and design-focused justification for the decision to group pescatarians with vegetarians and vegans. This decision may impair the value of the study for some researchers investigating interventions for meat-reduced and meat-abstaining diets due to the conflation of diet identities/behaviours.

We thank the reviewer for this comment. We appreciate the reviewer's comment that this might be of importance to researchers investigating interventions for meat-reduced and meat-abstaining diets referring to our study. As we have collected information for each dietary identity separately we are still able to report these identities separately. We have amended our protocol accordingly (P.11, L.41)